# Engineered Nanoparticles with Decoupled Photocatalysis and Wettability for Membrane-Based Desalination and Separation of Oil-Saline Water Mixtures

**DOI:** 10.3390/nano11061397

**Published:** 2021-05-25

**Authors:** Bishwash Shrestha, Mohammadamin Ezazi, Gibum Kwon

**Affiliations:** Department of Mechanical Engineering, University of Kansas, Lawrence, KS 66045, USA; bishwashs@ku.edu (B.S.); aminezazi@ku.edu (M.E.)

**Keywords:** photocatalytic nanoparticles, wettability, perfluorinated silica nanoparticles, oil–water separation, membrane fouling

## Abstract

Membrane-based separation technologies are the cornerstone of remediating unconventional water sources, including brackish and industrial or municipal wastewater, as they are relatively energy-efficient and versatile. However, membrane fouling by dissolved and suspended substances in the feed stream remains a primary challenge that currently prevents these membranes from being used in real practices. Thus, we directly address this challenge by applying a superhydrophilic and oleophobic coating to a commercial membrane surface which can be utilized to separate and desalinate an oil and saline water mixture, in addition to photocatalytically degrading the organic substances. We fabricated the photocatalytic membrane by coating a commercial membrane with an ultraviolet (UV) light-curable adhesive. Then, we sprayed it with a mixture of photocatalytic nitrogen-doped titania (N-TiO_2_) and perfluoro silane-grafted silica (F-SiO_2_) nanoparticles. The membrane was placed under a UV light, which resulted in a chemically heterogeneous surface with intercalating high and low surface energy regions (i.e., N-TiO_2_ and F-SiO_2_, respectively) that were securely bound to the commercial membrane surface. We demonstrated that the coated membrane could be utilized for continuous separation and desalination of an oil–saline water mixture and for simultaneous photocatalytic degradation of the organic substances adsorbed on the membrane surface upon visible light irradiation.

## 1. Introduction

Modulating the surface wettability plays a vital role in a solid-liquid system and has found increasing interest in practical applications, including self-cleaning [1,2], microfluidics [3,4], and liquid separation [5,6]. Based on the contact angle (*θ*) for high (e.g., water) and low surface tension liquids (e.g., oil), a surface can be grouped into four wettability classifications: omniphobic (*θ*_water_ > 90° and *θ*_oil_ > 90°), hydrophobic and oleophilic (*θ*_water_ > 90° and *θ*_oil_ < 90°), hydrophilic and oleophobic (*θ*_water_ < 90° and *θ*_oil_ > 90°), and omniphilic (*θ*_water_ < 90° and *θ*_oil_ < 90°) [7,8]. We [9,10,11] and others [12,13,14] have demonstrated that a low surface energy coating, in conjunction with the surface texture, can result in a non-wetting Cassie-Baxter state with air trapped between the contacting liquid and the solid surface.

Organofluorine is perhaps the most prevalent material for lowering the overall surface free energy and rendering the surface repellent to liquids [10]. It has been extensively employed to fabricate not only a surface with omniphobic [15,16] or superomniphobic wettability (i.e., *θ**_water_ > 150° and *θ**_oil_ > 150°) [9,17,18] but also those with selective (i.e., hydrophobic and oleophilic or hydrophilic and oleophobic) wettability [19,20,21,22]. For example, Mertens et al. [19] utilized a combination of photolithography and oxygen plasma treatment to fabricate a nanocrystalline diamond surface with hydrophobic and hydrophilic arrays. Howarter et al. [20] grafted a perfluorinated polyethylene glycol on a silicon surface which could exhibit hydrophilic and oleophobic wettability.

The utility of surfaces with selective wettability can be further extended by incorporating them with photocatalytic nanoparticles that can degrade organic substances upon light irradiation [23,24,25]. Such a photocatalytic surface with selective wettability has demonstrated the potential for a wide range of practical applications, including anti-fouling [26,27], self-cleaning [28,29], and bactericidal coating [30]. Recent studies [31,32,33,34] have revealed that membranes with selective wettability can further benefit by incorporating photocatalytic nanoparticles that can radically transform physical filtration into chemically reactive processes. Thus, it can progressively eliminate the inherent shortcomings of conventional membrane-based filtration, such as pollutant degradation and membrane fouling [23]. Zhao et al. [35] reported on a polyacrylonitrile membrane coated with a fluorinated agent and photocatalytic ZnO. The membrane exhibited remediation of wastewater upon ultraviolet (UV) light irradiation and resistance to fouling. Luster et al. [36] fabricated an N-doped, TiO_2_-coated alumina membrane and demonstrated the photocatalytic degradation of carbamazepine (CBZ) as a model pollutant under simulated solar irradiation. Further, Coelho et al. [26] coated a filter paper with zirconia-doped cerium to prepare a photocatalytic membrane. The membrane demonstrated flux recovery by photocatalytic degradation of humic acid as a foulant.

Before these photocatalytic membranes with desired wettability can be utilized for practical applications, they need to fulfill the following three conditions [37,38]. First, the membrane’s wettability must remain unchanged by a photocatalytic reaction. Similarly, its physical and chemical integrity must not be affected by a photocatalytic reaction, particularly if the membrane is organic. Lastly, the coating (e.g., photocatalytic nanoparticles) needs to remain undetached when a high hydraulic shear force is exerted.

In this work, we engineered a visible light-responsive photocatalytic coating with superhydrophilic and oleophobic wettability both in air and underwater by utilizing nitrogen-doped titania (N-TiO_2_) and perfluoro silane-grafted silica (F-SiO_2_) nanoparticles. The coating was sprayed onto a commercial membrane surface with a UV-curable adhesive. Subsequent irradiation of UV light resulted in a chemically heterogeneous surface with intercalating high surface energy (N-TiO_2_) and low surface energy (F-SiO_2_) regions that were securely bound to the surface. Both the wettability and the integrity of the membrane remained unaffected throughout the photocatalytic degradation process of the organic substances when exposed to visible light irradiation. This can be attributed to the robust chemistry of the cured adhesive that protected the perfluoro silane molecules grafted to the SiO_2_ nanoparticles, as well as the underlying membrane from the reactive radical species generated when it was exposed to visible light irradiation. Thus, the coated membrane can be utilized for continuous separation and desalination of an oil–saline water mixture and simultaneous photocatalytic degradation of the organic substances adsorbed on the membrane surface upon visible light irradiation.

## 2. Materials and Methods

### 2.1. Chemicals

Titanium butoxide (TBOT), tetraethyl orthosilicate (TEOS), triethylamine (TEA), sodium dodecyl sulfate (SDS), and sodium chloride (NaCl) were purchased from Millipore Sigma, St. Louis, MO, USA. The 1H,1H,2H,2H-perfluorodecyl trichlorosilane (perfluoro silane) was purchased from Alfa Aesar, Lancashire, UK. Ethanol, acetone, isopropyl alcohol, hydrochloric acid (HCl), nitric acid (HNO_3_), and n-hexadecane were purchased from Fisher Chemical, Fairlawn, NJ, USA. The Norland ultraviolet (UV) light-curable optical adhesive (NOA 61) was purchased from Norland Products Inc, Cranbury, NJ, USA. The commercial TRISEP ACM5 membrane was purchased from Sterlitech, Kent, WA, USA.

### 2.2. Synthesis of N-TiO_2_ Nanoparticles

Titanium butoxide (TBOT, 5.0 g) was added dropwise to isopropyl alcohol (79 g), followed by the addition of deionized (DI) water (910 g). Nitric acid (HNO_3_, 0.01 M) was added to the solution to adjust the pH to 2. Subsequently, triethylamine was added dropwise to the solution. Please note that the molar ratios of TEA to TBOT were 0.5, 1.0, 2.0, and 3.0. The solution was stirred for 12 h at 30 °C. The precipitates were collected by centrifugation and thoroughly rinsed by DI water and ethanol. Upon vacuum drying for 10 h, the deep-yellow, nitrogen-doped titanium dioxide (N-TiO_2_) nanoparticles were obtained.

### 2.3. Synthesis of F-SiO_2_ Nanoparticles

Tetraethyl orthosilicate (TEOS, 1.0 g) was added dropwise to a solution of hydrochloric acid (HCl, 0.01 M) in DI water (100 g), followed by the addition of 1H,1H,2H,2H-perfluorodecyl trichlorosilane (1.0 g). The solution was stirred for 60 min at 60 °C, and the precipitates were collected by centrifugation. Following a thorough rinsing by DI water and ethanol, and after 10 h of vacuum drying, the perfluoro silane-grafted silica (F-SiO_2_) nanoparticles were obtained.

### 2.4. Photocatalytic Membrane Fabrication

A commercial membrane surface was spin-coated with a Norland ultraviolet (UV) light-curable optical adhesive (NOA 61) (1.0 wt% in acetone). Please note that the commercial membrane (i.e., Sterlitech TRISEP ACM5) consisted of three layers of a thin polyamide layer with a molecular weight cut-off equal to 100 Da, a porous polysulfone layer, and a non-woven polyester as the support. The suspension of the N-TiO_2_ and F-SiO_2_ nanoparticles mixture (i.e., N-TiO_2_/F-SiO_2_) in DI water (10 wt%) was then sprayed (iWata spray gun, Anest iwata, Yokohama, Japan) onto the adhesive-coated membrane for one minute. The spraying distance and nitrogen gas pressure were maintained at 15 cm and 200 kPa, respectively. Please note that the concentrations of N-TiO_2_ nanoparticles in the N-TiO_2_/F-SiO_2_ mixture were 0 wt%, 20 wt%, 40 wt%, 60 wt%, 80 wt%, and 100 wt%. Subsequently, the membrane surface was irradiated by a long-wavelength UV light (100 W, λ = 365 nm, Analytikjena, Upland, CA, USA) for 5 min to cure the adhesive. The membrane was thoroughly rinsed with DI water and ethanol.

### 2.5. Preparation of the Oil-in-Water Emulsion Dissolved with Salt

An oil-in-water emulsion containing salt was prepared by vigorous mixing of n-hexadecane and DI water (10:90 vol:vol n-hexadecane:water) dissolved with salt (NaCl, 1.0 wt% with respect to the water mass). Sodium dodecyl sulfate (SDS, 0.015 wt%) was added to stabilize the emulsion.

### 2.6. N-TiO_2_ and F-SiO_2_ Size Measurements

The average size of the N-TiO_2_ and F-SiO_2_ nanoparticles was measured by utilizing dynamic light scattering (DLS) (ZetaPALS zeta potential analyzer, Brookhaven Instruments, Holtsville, NY, USA) equipped with a BI-9000AT digital autocorrelator. Suspensions of N-TiO_2_ (0.01 wt%) and F-SiO_2_ (0.01 wt%) were prepared in DI water, followed by ultrasonication.

### 2.7. N-TiO_2_ Crystal Structure Analyses

The crystal structure of N-TiO_2_ was studied by powder X-ray diffraction (XRD) (PANalytical Model X’Pert PRO diffractometer, PANalytical, Almelo, The Netherlands) with Cu Kα radiation (λ = 1.54 Å) by scanning at a rate of 2° (2*θ*) min^−1^. X-ray photoelectron spectroscopy (XPS) was utilized to study the nitrogen doping of N-TiO_2_. XPS was conducted by a Phi Versaprobe II, Ulvac-PHI, Kanagawa, Japan utilizing monochromatic source Mg Ka.

### 2.8. N-TiO_2_ and F-SiO_2_ Absorbance Measurements

Ultraviolet–visible (UV-Vis) spectrophotometry was utilized to analyze the nanoparticles’ absorbance spectrum. UV-Vis spectrophotometry was conducted by utilizing a Thermo Evolution 600, Waltham, MA, USA at a scan rate of 240 nm min^−1^ and a data interval of 2 nm.

### 2.9. N-TiO_2_ Photocatalytic Performance Measurements

The photocatalysis performance of N-TiO_2_ nanoparticles was analyzed by conducting the dye degradation test. UV-Vis spectrophotometry was utilized to study the dye degradation performance. N-TiO_2_ nanoparticles (0.5 wt%) were dispersed in DI water dissolved with Solvent Blue 38 dye (0.5 wt%). Of the dispersion, 20 mL was poured into a glass beaker equipped with a magnetic stirrer. The visible light (13.1 W, Sugarcube ultraLED, USHIO, Vergennes, VT, USA) guide was submerged into the beaker to irradiate the dispersion. A small quantity (2 mL) of the dispersion was taken every 1 h. It was centrifuged and filtered by filter paper, followed by UV-Vis spectrophotometry. UV-Vis spectrophotometry was conducted at a scan rate of 240 nm min^−1^ and a data interval of 2 nm.

### 2.10. Membrane Surface Analysis

Scanning electron microscopy (SEM, FEI Versa 3D DualBeam, Hillsboro, OR, USA) was utilized to study the surface porosity and texture of the N-TiO_2_/F-SiO_2_ coated membrane. An accelerating voltage of 10 kV was utilized. All surfaces were sputter-coated with a gold layer (≈4–5 nm) to prevent the charging effect.

### 2.11. Visible Light Intensity Measurement

A photometer (Fisherbrand Traceable DualDisplay Lightmeter, Control company, Webster, TX, USA) was utilized to measure the intensity of the incident visible light on the membrane surface. The photometer was placed underneath the top cover of the cross-flow cell and irradiated by the visible light source from the same distance (≈5 cm) at which the membrane was irradiated during the separation and desalination.

### 2.12. Salt and Oil Concentration Measurements

We determined the salt concentration in the water-rich permeate by calculating the electrical conductivity of the permeate and comparing the value with the calibration curve. Two probes (1 cm^2^ each) of a multimeter (GDT-3190, Gardner Bender, New Berlin, WI, USA) at a distance of 2 cm were submerged in the permeate (20 mL). The multimeter measured the electrical resistance (*R*), which was converted to electrical resistivity. Subsequently, the inverse of the electrical resistivity yielded the electrical conductivity (*s*). We determined the oil concentration by utilizing thermogravimetric analyses (TGA, PerkinElmer PYRIS 1, PerkinElmer, Waltham, MA, USA). Approximately 10 mg of the water-rich permeate was heated from room temperature (≈22 °C) to 110 °C at a rate of 5 °C min^−1^, followed by maintaining the temperature at 110 °C for 50 min.

### 2.13. Engineering a Continuous Separation and Desalination Apparatus

We engineered a continuous separation and desalination apparatus consisting of a cross-flow cell (CF042A, Sterlitech, Kent, WA, USA), a feed storage tank, a pump (2SF22SEEL, WEG industries, Jaraguá do Sul, Brazil), a differential pressure gauge (DPG409-500DWU, OMEGA, Stamford, CT, USA), a visible light source, and a permeate tank. The membrane with an effective surface area of ≈42 cm^2^ was sandwiched in between two transparent acrylic counterparts of the cross-flow cell. The membrane surface was irradiated by visible light of varying intensities.

## 3. Results and Discussion

### 3.1. Synthesis of N-TiO_2_ and Characterization of Its Photocatalysis upon Visible Light Irradiation

To fabricate a visible-light-responsive photocatalytic coating with selective wettability (i.e., hydrophilic and oleophobic), we utilized a mixture of nitrogen-doped titanium dioxide (N-TiO_2_) and perfluoro silane-grafted silica (F-SiO_2_) nanoparticles. N-TiO_2_ can degrade organic substances when it is exposed to visible light irradiation [39] and exhibit hydrophilic wettability [40], while F-SiO_2_ can lower the overall surface free energy (γ_sv_) [41]. We hypothesized that an optimal balance of N-TiO_2_ and F-SiO_2_ could result in hydrophilic and oleophobic wettability.

To verify this hypothesis, we first synthesized the N-TiO_2_ nanoparticles by utilizing the sol–gel method [42] (further described in the Section 2). Triethylamine (TEA) and titanium butoxide (TBOT) were used as a nitrogen dopant and a TiO_2_ precursor, respectively (Figure 1a). Hydrolysis of the TBOT (concentration = 0.5 wt%) in an acidic solution (pH ≈ 2.0) that was dissolved with TEA (concentration = 0.3 wt%) resulted in N-TiO_2_ nanoparticles with an average diameter of 50 nm ± 1 nm (Appendix A). During this reaction, the TEA introduced nitrogen (e.g., elemental nitrogen or complex nitrogen species such as NO, NO_2_, or NH) into the TiO_2_ lattice [43]. X-ray photoelectron spectroscopy (XPS) spectra demonstrated a peak at a binding energy of ≈399.2 eV, which indicated the presence of anionic nitrogen from the O-Ti-N bond [42], while the neat TiO_2_ lacked such a peak (Figure 1b).

Doping with nitrogen can narrow the bandgap energy of the TiO_2_, which may extend the absorption spectra further toward the visible light region (i.e., 390 nm < λ < 750 nm) [44]. The ultraviolet–visible (UV-Vis) spectrophotometry data verified that N-TiO_2_ absorbed a broad range of the visible light spectrum, whereas the neat TiO_2_ and SiO_2_ showed negligible absorption (Figure 1c). Furthermore, the results showed that the N-TiO_2_ fabricated with a higher molar ratio of TEA to TBOT (i.e., higher dopant concentration) tended to exhibit stronger absorbance in the range of wavelengths from 390 nm to 750 nm. However, once the molar ratio exceeded 2.0, this would have a negligible effect on the absorbance of the resulting N-TiO_2_. Thus, we utilized a TEA-to-TBOT molar ratio of 2.0 (hereafter denoted as N-TiO_2_) in this study.

Given that the resulting N-TiO_2_ exhibited a photocatalytic anatase crystalline phase (Appendix A), we demonstrated that it could degrade organic substances upon visible light irradiation. Figure 1d presents the time-dependent degeneration of the absorbance for water dissolved with organic dye (Solvent Blue 38, concentration = 0.5 wt%) and N-TiO_2_ (concentration = 0.5 wt%). The light intensity (*I*) was maintained at *I* ≈ 198 mW cm^−2^ (see Section 2). The water solution became nearly colorless after 10 h of irradiation, indicating that almost all of the dye molecules were degraded. Please note that the dye molecules degraded more rapidly when placed under a higher intensity of light (Appendix A).

The F-SiO_2_ nanoparticles were synthesized through the hydrolysis of tetraethyl orthosilicate (TEOS), followed by grafting of 1H,1H,2H,2H-perfluorodecyl trichlorosilane [41,45] (see Section 2). The average diameter of the F-SiO_2_ nanoparticles was 50 nm ± 2 nm (Appendix A).

### 3.2. Fabrication and Characterization of N-TiO_2_/F-SiO_2_ Coated Membrane

By utilizing N-TiO_2_ and F-SiO_2_ nanoparticles, we created a photocatalytic membrane with hydrophilic and oleophobic wettability both in air and saline water. A commercial filter (TRISEP ACM5) was utilized as a membrane, on which our visible light-responsive photocatalytic coating with selective wettability was applied. Please note that the TRISEP ACM5 was chosen because of its applicability to a wide range of separation processes [46]. First, we spin-coated the membrane surface with a thiol-ene-based UV-curable adhesive. Immediately following this step, a solution of N-TiO_2_ and F-SiO_2_ (i.e., N-TiO_2_/F-SiO_2_) (concentration = 10 wt%, Section 2) was sprayed for one minute. Then, the membrane was irradiated by UV light (λ = 365 nm, intensity ≈ 78 mW cm^−2^) for five minutes at room temperature (≈22 °C) to cure the adhesive. Please note that we varied the concentrations of N-TiO_2_/F-SiO_2_ (e.g., 0, 20 wt%, 40 wt%, 60 wt%, 80 wt%, and 100 wt%). Figure 2 shows a schematic illustrating the overall process of membrane fabrication.

The resulting membrane’s surface was covered with a N-TiO_2_/F-SiO_2_ coating possessing a hierarchical roughness with a re-entrant texture (Figure 3a and the inset image). We measured the advancing (*θ**_adv_) and receding (*θ**_rec_) contact angles for the saline water (1.0 wt% NaCl in DI water, γ_lv_ = 73.1 mN m^−1^) and oil (n-hexadecane, γ_lv_ = 27.5 mN m^−1^) in air (Figure 3b). The results indicated that a membrane coated with N-TiO_2_/F-SiO_2_ with a lower N-TiO_2_ concentration exhibited higher contact angles for both saline water and oil. We found that when the N-TiO_2_ concentration reached 60 wt%, the contact angle for saline water became zero (i.e., *θ**_saline water, adv (in-air)_ = 0° and *θ**_saline water, rec (in-air)_ = 0°), while that for oil remained constant (*θ**_oil, adv (in-air)_ = 95° ± 4°, *θ**_oil, rec (in-air)_ = 61° ± 3°). Further increases in the N-TiO_2_ concentrations resulted in sharp decreases in contact angles for oil.

Given that membrane operations in real-world applications often result in continuous immersion in liquids (e.g., water), we also measured the contact angles for oil on the membrane surface submerged in saline water (Figure 3c). The results indicated that a membrane with a lower in-air water contact angle was likely to have a higher oil contact angle when submerged in saline water. For example, a membrane coated with N-TiO_2_/F-SiO_2_ of 60 wt% N-TiO_2_ (i.e., N-TiO_2_/F-SiO_2_ (60 wt%)) exhibited contact angles of *θ**_oil, adv (under saline water)_ = 175° ± 3° and *θ**_oil, rec (under saline water)_ = 171° ± 2°, while the one coated with N-TiO_2_/F-SiO_2_ (20 wt%) exhibited *θ**_oil, adv (under saline water)_ = 169° ± 2° and *θ**_oil, rec (under saline water)_ = 161° ± 3°. The results could be further corroborated by analyzing the adhesion force of an oil droplet (n-hexadecane) placed on the membrane surface that was submerged in saline water [47] (Figure 3c). The adhesion force was 1.1 ± 0.3 mN on a membrane coated with N-TiO_2_/F-SiO_2_ (60 wt%) and 1.7 ± 0.4 mN on a membrane coated with N-TiO_2_/F-SiO_2_ (20 wt%).

The membrane’s wettability for saline water can affect the flux because it determines the breakthrough pressure (i.e., the maximum pressure difference across the membrane that is required for a liquid to permeate through it) [48]. We measured the flux of saline water (1.0 wt% NaCl in DI water) through the membranes coated with various concentrations of N-TiO_2_/F-SiO_2_. A total of 100 L of saline water was continuously fed for 180 min through the membrane, which was attached to a cross-flow cell. We measured the volume of the permeate every 10 min. It must be noted that the transmembrane pressure (TMP) (i.e., the pressure exerted across the membrane) was maintained at ∆*p* ≈ 760 kPa ± 9 kPa. Figure 3d shows the normalized flux (*J*_n_) of the permeate, which is defined as *J*_t_/*J*_o_, where *J*_t_ = ∆*m*/A*ρ*∆*t*∆*p*. Here, ∆*m* represents the change in the permeate mass during a particular time interval (i.e., ∆*t =* 10 min), *A* is the membrane’s projected area (*A* ≈ 42 cm^2^), *ρ* is the density of the permeate (*ρ* = 1000 kg m^−3^), and *J*_o_ symbolizes the flux over the TMP value obtained during the first three minutes of submersion. While *J*_n_ gradually decreased and reached a constant value (*J*_n_ ≈ 0.59 ± 0.04) after approximately *t* = 150 min for all membranes, one coated with N-TiO_2_/F-SiO_2_ with a higher concentration of N-TiO_2_ showed less of a decrease in the *J*_n_ value at *t* = 150 min. For example, a membrane coated with N-TiO_2_/F-SiO_2_ (60 wt%) yielded a *J*_n_ value of 0.61, whereas one coated with N-TiO_2_/F-SiO_2_ (20 wt%) showed a value of 0.56 at *t* = 150 min. This finding can be attributed to the difference in the breakthrough pressure of saline water [48] (Appendix A). Please note that the as-purchased filter and the one coated only with the cured adhesive exhibited *J*_n_ = 0.59 and *J*_n_ = 0.58, respectively, at *t* = 150 min (Figure 3d). This clearly indicates that neither the cured adhesive nor the N-TiO_2_/F-SiO_2_ coating affected the membrane’s flux. We then measured the salt rejection (*ξ*) of the membranes coated with various concentrations of N-TiO_2_/F-SiO_2_ (Appendix A). We also observed almost no change in the mass of the membranes after 180 min, implying that the N-TiO_2_/F-SiO_2_ nanoparticles were securely bound to the membrane surface (Appendix A).

### 3.3. Continuous Separation and Desalination of an Oil–Saline Water Mixture and Simultaneous Photocatalytic Degradation of Organic Foulants upon Visible Light Irradiation

The photocatalytic capability of our coating, in conjunction with its hydrophilic and oleophobic wettability both in air and under saline water, enables the membrane to separate and desalinate an oil–saline water mixture while simultaneously degrading the organic foulants adsorbed onto the membrane surface during exposure to visible light irradiation [49,50]. To demonstrate this, we mounted the membrane to an apparatus and irradiated it with visible light (Figure 4a). The feed oil–saline water mixture was continuously fed to the cell while the water-rich permeate continuously passed through the membrane and collected in a permeate tank. Here, we utilized an emulsion of n-hexadecane in water (10:90 vol:vol, n-hexadecane:water) dissolved with salt (1.0 wt% NaCl with respect to the water mass) that was stabilized by sodium dodecyl sulfate (SDS) (see Section 2). We tested membranes coated with various concentrations of N-TiO_2_/SiO_2_. Of note, all membranes were prewetted by soaking them in saline water (1.0 wt% NaCl) for 150 min to obtain a constant flux over the TMP (*J*_prewet_) before being subjected to the feed emulsion.

When the feed emulsion was introduced, the flux of the water-rich permeate began to rapidly decrease due to membrane fouling by the oil (Figure 4b and Table 1). This is a consequence of the oil’s adsorption to the membrane surface, which can hamper the permeation of water and cause a rapid decline in the flux. Note that *J*_n_ is defined as *J*_t_/*J*_prewet_. According to our findings, an N-TiO_2_/F-SiO_2_-coated membrane with a higher N-TiO_2_ composition exhibited a steeper decrease in flux. For example, when a membrane was coated with N-TiO_2_/F-SiO_2_ (80 wt%), *J*_n_ ≈ 0.17 (*J*_t_ ≈ 0.0135 Lm^−2^h^−1^kPa^−1^). However, when it was coated with N-TiO_2_/F-SiO_2_ (20 wt%), *J*_n_ ≈ 0.30 (*J*_t_ ≈ 0.0163 Lm^−2^h^−1^kPa^−1^) at *t* = 180 min. This can be primarily attributed to the fact that N-TiO_2_ is more vulnerable to oil adsorption [51], while F-SiO_2_ can repel it [52] (Appendix A). Note that such a rapid flux decline in the membranes after the introduction of an oil–water mixture has been reported previously [53,54]. Despite the membranes’ decline in flux, the concentration of oil in the permeate remained very low (i.e., <0.1 wt%; see Appendix A).

When the water-rich permeate flux reached a constant value at *t* = 180 min, we began to irradiate the membrane surface with visible light (*I* ≈ 198 mW cm^−2^) to induce photocatalytic degradation of the surface’s adsorbed oil. This was a result of electron–hole (e^−^-h^+^) pairs generated upon light irradiation with an energy greater than the bandgap energy of the photocatalyst (e.g., N-TiO_2_) [55]. The electrons and holes can react with the ambient molecules (e.g., oxygen or water) and generate reactive radicals such as hydroxyl, which can remove organic contaminants such as oil by chemical oxidation (or reduction) [31]. The membrane was then continuously subjected to a fresh feed emulsion. We observed that a membrane coated with N-TiO_2_/F-SiO_2_ with a higher concentration of N-TiO_2_ caused a more significant increase in the *J*_n_ value, which was a direct result of the photocatalysis-driven recovery of the clean membrane surface that exhibited a lower breakthrough pressure for saline water (Figure 4c and Table 2). For example, a membrane coated with N-TiO_2_/F-SiO_2_ (80 wt%) yielded *J*_n_ ≈ 0.24 (*J*_t_ ≈ 0.0190 Lm^−2^h^−1^kPa^−1^), while the one coated with N-TiO_2_/F-SiO_2_ (20 wt%) exhibited *J*_n_ ≈ 0.31 (*J*_t_ ≈ 0.0174 Lm^−2^h^−1^kPa^−1^) after 60 min of irradiation.

While an in situ photocatalysis-driven recovery of the clean membrane surface is presented in this work, a majority of the previous studies have demonstrated this ex situ. For example, Zhang et al. [33] synthesized an electrospun membrane anchored with photocatalytic β-FeOOH nanorods. The membrane demonstrated that it could photocatalytically degrade the surface adsorbed organic matter and recover its flux upon visible light irradiation after 40 min. Peyravi et al. [56] incorporated photocatalytic TiO_2_/zeolite into a composite membrane and demonstrated ≈83.6% flux recovery under UV irradiation. Liu et al. [34] fabricated a membrane utilizing TiO_2_/carbon nitride nanosheets and showed membrane surface cleaning with a flux recovery ratio of >95%. In addition, Kovács et al. [57] demonstrated that a TiO_2_-coated ultrafiltration membrane could almost completely recover its original flux after 1 h of UV irradiation. Furthermore, Xie et al. [58] fabricated a photocatalytic membrane using β-FeOOH, which demonstrated >98% flux recovery within 10 min under visible light.

We also measured the contact angles for oil and saline water after 60 min of visible light irradiation and found that they remained unchanged (Figure 4d). This indicates that the photocatalytic reaction taking place within the N-TiO_2_ did not affect the perfluoro silane molecules grafted to the SiO_2_ (Appendix A). Moreover, both the salt rejection (*ξ*) of the membranes and the cured adhesive layer remained nearly unchanged (Appendix A).

## 4. Conclusions

In summary, we have developed a photocatalytic coating with hydrophilic and oleophobic wettability by intercalating a mixture of visible light-responsive N-TiO_2_ and low surface energy F-SiO_2_ nanoparticles. We tested the feasibility of our membrane coating by spraying it on a commercial membrane surface with UV-curable adhesive. Subsequent irradiation with UV light resulted in a chemically heterogeneous surface with intercalating high surface energy (N-TiO_2_) and low surface energy (F-SiO_2_) regions that were securely bound to the surface. Our membrane could recover the flux upon visible light irradiation. We attributed this to the photocatalytic degradation of the surface adsorbed oil when placed under visible light irradiation. Such photocatalytic degradation did not compromise the wettability or integrity of the membrane due to the robust chemistry of the adhesive. We engineered an apparatus that enabled the continuous separation and desalination of a surfactant-stabilized oil-in-water emulsion that was dissolved with salt and the photocatalytic degradation of organic substances that were adsorbed on the coated membrane surface when it was exposed to visible light irradiation. It was found that the coated membrane was able to recover its permeate flux in situ when placed under visible light irradiation. We envision that our membrane will have a wide range of practical applications, including wastewater treatment, fuel purification, and desalinating brackish water.

## Figures and Tables

**Figure 1 nanomaterials-11-01397-f001:**
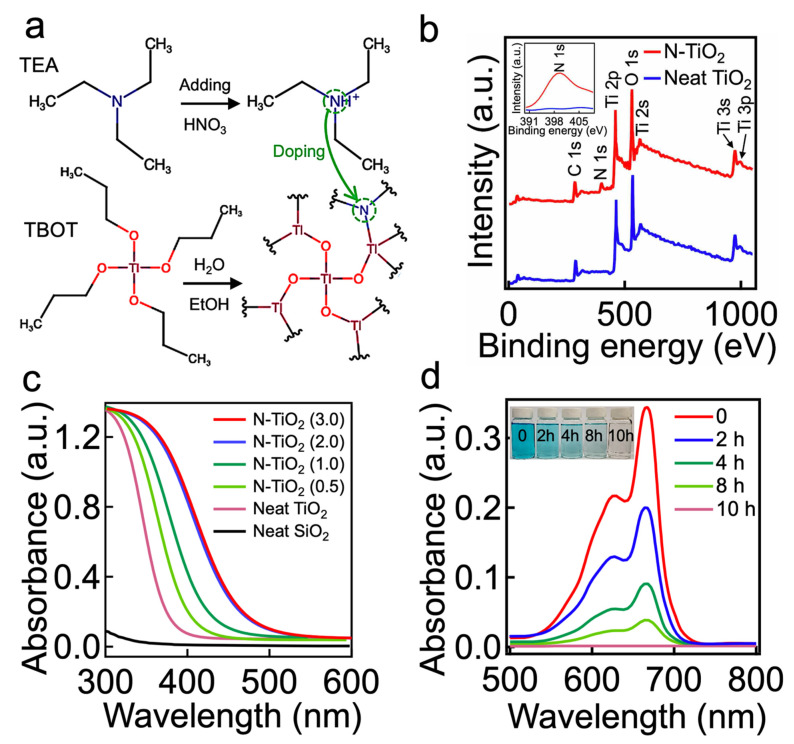
(**a**) Schematic illustrating the synthesis of N-TiO_2_ nanoparticles by utilizing titanium butoxide (TBOT) and triethylamine (TEA) as a TiO_2_ precursor and nitrogen dopant, respectively. (**b**) The XPS spectrum of N-TiO_2_, exhibiting characteristic peaks of N 1s, Ti 2p, and O 1s. The spectrum of a neat TiO_2_ is also shown for comparison. The inset shows the core level spectrum of characteristic N 1s. (**c**) Ultraviolet–visible (UV-Vis) absorption spectra of N-TiO_2_ synthesized by varied molar ratios of TEA to TBOT (e.g., 0.5, 1.0, 2.0, and 3.0). Neat TiO_2_ and SiO_2_ absorption spectra are also shown for comparison. (**d**) UV-Vis absorption spectra of water dissolved with N-TiO_2_ and Solvent Blue dye as a function of the visible light irradiation time. The inset is a photograph showing the water dissolved with Solvent Blue dye after visible light irradiation for 2 h, 4 h, 8 h, and 10 h. The as-prepared water dissolved with Solvent Blue dye (concentration = 0.5 wt%) is also shown.

**Figure 2 nanomaterials-11-01397-f002:**
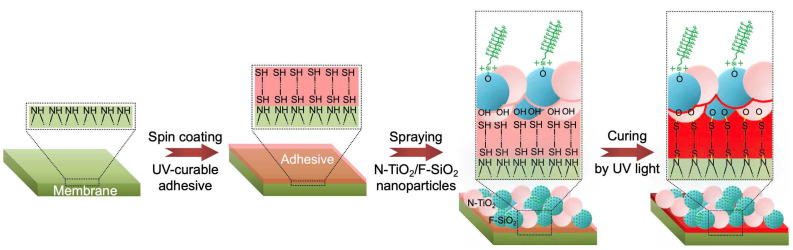
Schematic illustrating the fabrication of a photocatalytic membrane with hydrophilic and oleophobic wettability. A commercial filter is coated with an ultraviolet (UV) light-curable adhesive, followed by being sprayed with a mixture of N-TiO_2_ and F-SiO_2_ nanoparticles. The membrane was placed under UV light for curing.

**Figure 3 nanomaterials-11-01397-f003:**
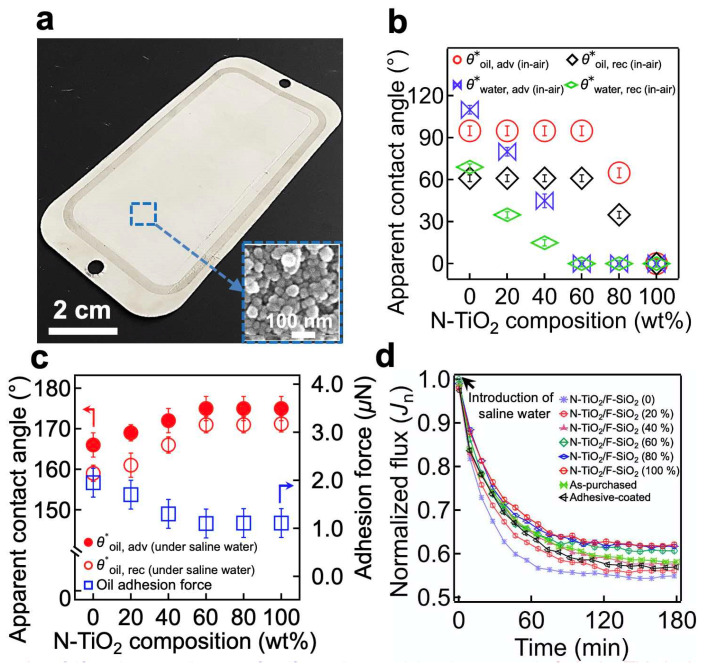
(**a**) A photograph showing a commercial filter coated with N-TiO_2_/F-SiO_2_ (60 wt%). The inset displays a scanning electron microscopy (SEM) image of the membrane surface, showing a hierarchical roughness with a re-entrant texture. (**b**) The measured in-air advancing and receding apparent contact angles for saline water (1.0 wt% NaCl in DI water) and oil (n-hexadecane) on the membrane surface, coated with N-TiO_2_/F-SiO_2_ with varied compositions. (**c**) The measured advancing and receding apparent contact angles, as well as the adhesion force for an oil droplet (n-hexadecane) on the membrane surface, coated with N-TiO_2_/F-SiO_2_ with various compositions submerged in saline water. (**d**) The normalized flux (*J*_n_) of the permeate through the membranes coated with N-TiO_2_/F-SiO_2_ with varied compositions. The data for the as-purchased commercial filter and for the one coated only with cured adhesive were also provided for comparison.

**Figure 4 nanomaterials-11-01397-f004:**
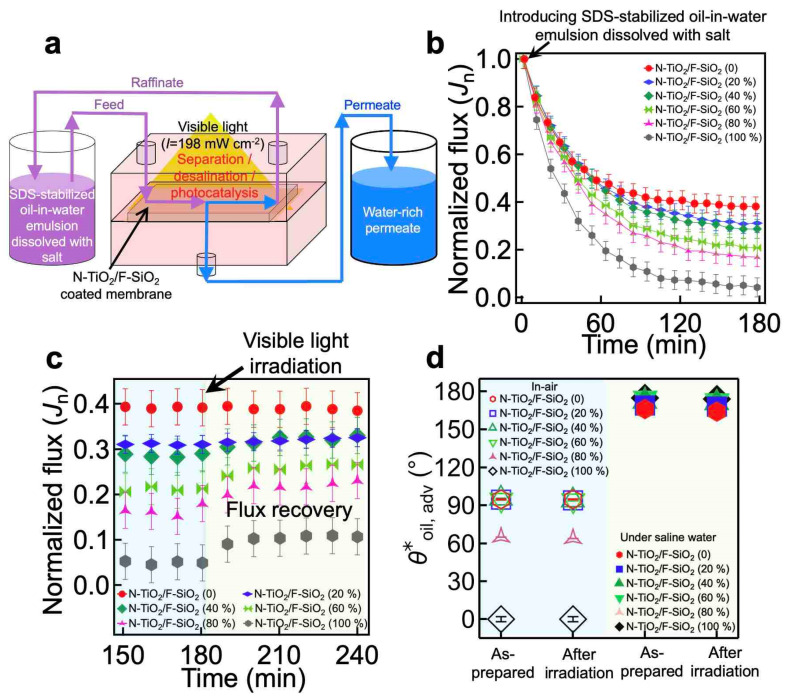
(**a**) Schematic illustrating the cross-flow apparatus that enables continuous separation and desalination of an oil–saline water mixture and simultaneous photocatalytic degradation of the organic foulants adsorbed onto the membrane’s surface upon visible light irradiation. (**b**) The normalized flux (*J*_n_) of the water-rich permeate through the membranes coated with N-TiO_2_/F-SiO_2_ with varied compositions that are subjected to sodium dodecyl sulfate (SDS)-stabilized n-hexadecane-in-water emulsion (10:90 vol:vol, n-hexadecane:water) dissolved with salt (1.0 wt% NaCl with respect to the water mass). (**c**) The normalized flux (*J*_n_) of the water-rich permeate through the membranes being irradiated by visible light (*I* ≈ 198 mW cm^−2^). (**d**) The apparent contact angles for oil (n-hexadecane) droplets on the membranes in air and under saline water before and after visible light irradiation for 120 min. The data of the as-prepared membranes are also shown for comparison.

**Table 1 nanomaterials-11-01397-t001:** The values of the flux of the water-rich permeate through the membranes coated with N-TiO_2_/F-SiO_2_ with varied compositions that were subjected to sodium dodecyl sulfate (SDS)-stabilized n-hexadecane-in-water emulsion (10:90 vol:vol, n-hexadecane:water) dissolved with salt (1.0 wt% NaCl with respect to the water mass).

N-TiO_2_/F-SiO_2_ Compositions	Flux at t = 0 (*J*_o_) (Lm^−2^h^−1^kPa^−1^)	Flux at t = 180 min (*J*_t_) (Lm^−2^h^−1^kPa^−1^)
N-TiO_2_/F-SiO_2_ (0)	0.0467	0.0177
N-TiO_2_/F-SiO_2_ (20 wt%)	0.0547	0.0163
N-TiO_2_/F-SiO_2_ (40 wt%)	0.0627	0.0174
N-TiO_2_/F-SiO_2_ (60 wt%)	0.0786	0.0157
N-TiO_2_/F-SiO_2_ (80 wt%)	0.0826	0.0135
N-TiO_2_/F-SiO_2_ (100 wt%)	0.0826	0.0033

**Table 2 nanomaterials-11-01397-t002:** The values of the flux of the water-rich permeate through the membranes coated with N-TiO_2_/F-SiO_2_ with varied compositions after 60 min of irradiation with visible light (*I* ≈ 198 mW cm^−2^).

N-TiO_2_/F-SiO_2_ Compositions	Flux at t = 180 min (before Irradiation) (Lm^−2^h^−1^kPa^−1^)	Flux at t = 240 min (after 60 min of Irradiation) (Lm^−2^h^−1^kPa^−1^)
N-TiO_2_/F-SiO_2_ (0)	0.0177	0.0177
N-TiO_2_/F-SiO_2_ (20 wt%)	0.0163	0.0174
N-TiO_2_/F-SiO_2_ (40 wt%)	0.0174	0.0200
N-TiO_2_/F-SiO_2_ (60 wt%)	0.0157	0.0212
N-TiO_2_/F-SiO_2_ (80 wt%)	0.0135	0.0190
N-TiO_2_/F-SiO_2_ (100 wt%)	0.0033	0.0090

## Data Availability

The datasets generated during and/or analyzed during this study are not publicly available but are available from the corresponding author on reasonable request.

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
