# Peer review of "Engineered Nanoparticles with Decoupled Photocatalysis and Wettability for Membrane-Based Desalination and Separation of Oil-Saline Water Mixtures"

_nanomaterials, 2021, doi:10.3390/nano11061397_

Round 1

Reviewer 1 Report

The manuscript is interesting and well written. Synthesis and characterization of composite membranes are well described and many details about their properties are shown and discussed, both in the manuscript and in the supporting info.

In partial contrast to the premises, the part concerning the test is a little less significant and the results seem to suggest some possible problems in the application of this technology to solutions rich in organic substances and more difficult to degrade. Nonetheless, in my opinion the work is worthy of publication after addressing a few issues listed below.

Remarks:

  • Section 2.4: Some more info should be added about the “commercial membrane” used for the experiments
  • Section 3.1, results shown in Fig 1d: Did the authors check the decolorization of a control system and have they verified the possibility of adsorption of the dye on the membrane?
  • Section 3.2: The authors wrote “The resulting membrane remains unchanged after spraying…”; please, can they explain what's the meaning of "unchanged" in this contest? 
  • Figure 3.b: Many symbols in the figure are overlapped and difficult to attribute to the correct set of data. I think the authors should modify the graphic to make it clearer to readers.
  • Section 4: The authors wrote about “… the photocatalytic degradation of organic substances …”, but they did not discuss or show data about this supposed degradation, except the enhance of flux recovery after irradiation. Please add more details to the manuscript or modify this consideration.

Reviewer 2 Report

In this work, a photocatalytic coating based on N-TiO2 nanoparticles (visible light-responsive) and FSiO2 nanoparticles is developed and (low surface energy). Spraying this coating on a commercial membrane allows proving that it is possible to use the coating for separation purposes, i.e. oil and
saline water separation from a mixture. It is shown that this procedure is possible to be applied in a continuous mode to separate oil and saline water under visible light exposure.

This work is well written, given a proper background in the introduction, all the necessary details in the experimental part, and the figures shown are quite good to understand the work. However, it would be advisable to strengthen the discussion in section 3, particularly at the time to explain the mechanism and to address the recovery of the permeable flux. It would be advisable to include a brief comparison with the recent results in the subject, so that the advantage of the proposed approach can be better
valued.

Another point to be addressed is the fact that in the conclusions it mentions it is expected to get a broad range of application for the proposed method, and among the expected application wastewater treatment, fuel purification, and so on are mentioned. In this regard, this application requires the
utilization of a method able to process large amounts. How do you evaluate the cost-practical feasibility when one of the reactants to synthesize the photocatalytic coating ranges in about 100 USD each 10 g (1H,1H,2H,2H-perfluorodecyl trichlorosilane). Despite the fact that 1 g is used, would be realistic to scale up the method for such application? Or it is better to look for lab-middle scale applications?

Reviewer 3 Report

This manuscript developed a photocatalytic coating with hydrophilic and oleophobic wettability by intercalating a mixture of visible light-responsive N-TiO2 and low surface energy F-SiO2 nanoparticles. In addition, an apparatus was engineered which enabled the continuous separation and desalination of a surfactant-stabilized oil-in-water emulsion and the photocatalytic degradation of organic substances under visible light irradiation. The present study is interesting, and it is potentially to be widely used in wastewater treatment, fuel purification, desalinating brackish water. It is suggested that some papers closely related to the present study can be referenced such as Appl. Catal. B: Environ. 2019, 249, 1-8 and Chin. J. Catal. 2020, 41, 1451-1467.
